# A Syntactic Neural Model for General-Purpose Code Generation

## Abstract

We consider the problem of parsing natural language descriptions into source code written in a general-purpose programming language like Python. Existing data-driven methods treat this problem as a language generation task without considering the underlying syntax of the target programming language. Informed by previous work in semantic parsing, in this paper we propose a novel neural architecture powered by a grammar model to explicitly capture the target syntax as prior knowledge. Experiments find this an effective way to scale up to generation of complex programs from natural language descriptions, achieving state-of-the-art results that well outperform previous code generation and semantic parsing approaches.

## 1 Introduction

Every programmer has experienced the situation where they know what they want to do, but do not have the ability to turn it into a concrete implementation. For example, a Python programmer may want to "*sort my_list in descending order*," but not be able to come up with the proper syntax `sorted(my_list, reverse=True)` to realize his intention. To resolve this impasse, it is common for programmers to search the web in natural language (NL), find an answer, and modify it into the desired form (Brandt et al., 2009, 2010). However, this is time-consuming, and thus the software engineering literature is ripe with methods to directly generate code from NL descriptions, mostly with hand-engineered methods highly tailored to specific programming languages (Balzer, 1985; Little and Miller, 2009; Gvero and Kuncak, 2015).

In parallel, the NLP community has developed methods for data-driven semantic parsing, which attempt to map NL to structured logical forms executable by computers. These logical forms can be general-purpose meaning representations (Clark and Curran, 2007; Banarescu et al., 2013), formalisms for querying knowledge bases (Tang and Mooney, 2001; Zettlemoyer and Collins, 2005; Berant et al., 2013) and instructions for robots or personal assistants (Artzi and Zettlemoyer, 2013; Quirk et al., 2015), among others. While these methods have the advantage of being learnable from data, compared to the programming languages (PLs) in use by programmers, the *domain-specific* languages targeted by these works have a schema and syntax that is relatively simple.

Recently, Ling et al. (2016) have proposed a data-driven code generation method for high-level, *general-purpose* PLs like Python and Java. This work treats code generation as a sequence-to-sequence modeling problem, and introduce methods to generate words from character-level models, and copy variable names from input descriptions. However, unlike most work in semantic parsing, it does not consider the fact that code has to be well-defined programs in the target syntax.

In this work, we propose a data-driven syntax-based neural network model tailored for generation of general-purpose PLs like Python. In order to capture the strong underlying syntax of the PL, we define a model that transduces an NL statement into an Abstract Syntax Tree (AST; Fig. 1(a), § 2) for the target PL. ASTs can be deterministically generated for all well-formed programs using standard parsers provided by the PL, and thus give us a way to obtain syntax information with minimal engineering. Once we generate an AST, we can use deterministic generation tools to convert the AST into surface code. We hypothesize that such a structured approach has two benefits.

| Production Rule | Role | Explanation |
|---|---|---|
| Call ↦ expr[*func*] expr*[*args*] keyword*[*keywords*] | Function Call | ▷ *func*: the function to be invoked ▷ *args*: arguments list ▷ *keywords*: keyword arguments list |
| If ↦ expr[*test*] stmt*[*body*] stmt*[*orelse*] | If Statement | ▷ *test*: condition expression ▷ *body*: statements inside the If clause ▷ *orelse*: elif or else statements |
| For ↦ expr[*target*] expr*[*iter*] stmt*[*body*] stmt*[*orelse*] | For Loop | ▷ *target*: iteration variable ▷ *iter*: enumerable to iterate over ▷ *body*: loop body ▷ *orelse*: else statements |
| FunctionDef ↦ identifier[*name*] arguments*[*args*] stmt*[*body*] | Function Def. | ▷ *name*: function name ▷ *args*: function arguments ▷ *body*: function body |

Table 1: Example production rules for common Python statements (Python Software Foundation, 2016)

First, we hypothesize that structure can be used to constrain our hypothesis space, ensuring generation of well-formed code. To this end, we propose a syntax-driven neural code generation model. The backbone of our approach is a *grammar model* (§ 3) which formalizes the generation story of a derivation AST into sequential application of *actions* that either apply production rules (§ 3.1), or emit terminal tokens (§ 3.2). The underlying syntax is therefore encoded in the grammar model *a priori* as the set of possible actions. Our approach frees the model from recovering the underlying grammar from limited training data, and instead enables the system to focus on learning the compositionality among existing grammar rules. Xiao et al. (2016) have noted that this imposition of structure on neural models is useful for semantic parsing, and we expect this to be even more important for general-purpose PLs where the syntax trees are larger and more complex.

Second, we hypothesize that structural information helps to model information flow within the network, which naturally reflects the recursive structure of PLs. To test this, we extend a standard recurrent neural network (RNN) decoder to allow for additional neural connections which reflect the recursive structure of an AST (§ 4.2). As an example, when expanding the node ⋆ in Fig. 1(a), we make use of the information from both its parent and left sibling (the dashed rectangle). This enables us to locally pass information of relevant code segments via neural network connections, resulting in more confident predictions.

Experiments (§ 5) on two Python code generation tasks show 11.7% and 9.3% absolute improvements in accuracy against the state-of-the-art system (Ling et al., 2016). Our model also gives competitive performance on a standard semantic parsing benchmark.

## 2 The Code Generation Problem

Given an NL description $x$, our task is to generate the code snippet $c$ in a modern PL based on the in-

tent of $x$. We attack this problem by first generating the underlying AST. We define a probabilistic grammar model of generating an AST $y$ given $x$: $p(y|x)$. The best-possible AST $\hat{y}$ is then given by

$$\hat{y} = \arg\max_y p(y|x). \qquad (1)$$

$\hat{y}$ is then deterministically converted to the corresponding surface code $c$.[1] While this paper uses examples from Python code, our method is PL-agnostic.

Before detailing our approach, we first present a brief introduction of the Python AST and its underlying grammar. The Python abstract grammar contains a set of production rules, and an AST is generated by applying several production rules composed of a head node and multiple child nodes. For instance, the first rule in Tab. 1 is used to generate the function call sorted(·) in Fig. 1(a). It consists of a head node of type Call, and three child nodes of type expr, expr* and keyword*, respectively. Labels of each node are noted within brackets. In an AST, non-terminal nodes sketch the general structure of the target code, while terminal nodes can be categorized into two types: *operation terminals* and *variable terminals*. Operation terminals correspond to basic arithmetic operations like AddOp.Variable terminal nodes store values for variables and constants of built-in data types[2]. For instance, all terminal nodes in Fig. 1(a) are variable terminal nodes.

## 3 Grammar Model

Before detailing our neural code generation method, we first introduce the grammar model at its core. Our probabilistic grammar model defines the generative story of a derivation AST. We factorize the generation process of an AST into sequential application of *actions* of two types:

- APPLYRULE[*r*] applies a production rule $r$ to the current derivation tree;

---

[1]We use astor library to convert ASTs into Python code.
[2]bool, float, int, str.

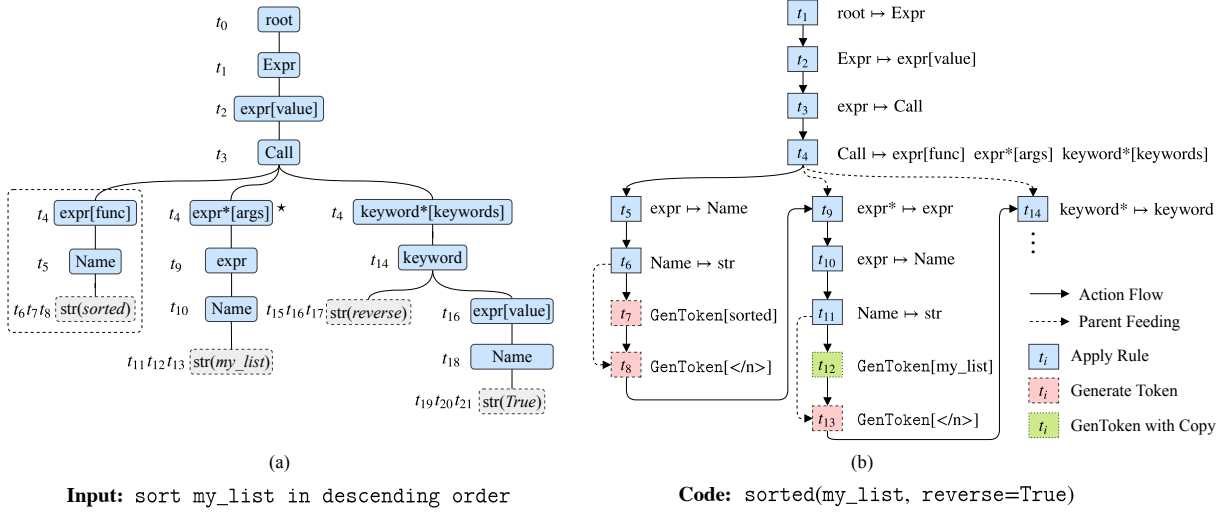

(a)

**Input:** `sort my_list in descending order`

(b)

**Code:** `sorted(my_list, reverse=True)`

Figure 1: (a) the Abstract Syntax Tree (AST) for the given example code. Dashed nodes denote terminals. Nodes are labeled with time steps during which they are generated. (b) the action sequence (up to $t_{14}$) used to generate the AST in (a)

- GENTOKEN$[v]$ populates a variable terminal node by appending a terminal token $v$.

Fig. 1(b) shows the generation process of the target AST in Fig. 1(a). Each node in Fig. 1(b) indicates an action. Action nodes are connected by solid arrows which depict the chronological order of the action flow. The generation proceeds in depth-first, left-to-right order (dotted arrows represent parent feeding, explained in § 4.2.1).

Formally, under our grammar model, the probability of generating an AST $y$ is factorized as:

$$p(y|x) = \prod_{t=1}^{T} p(a_t|x, a_{<t}), \qquad (2)$$

where $a_t$ is the action taken at time step $t$, and $a_{<t}$ is the sequence of actions before $t$. We will explain how to compute Eq. (2) in § 4. Put simply, the generation process begins from a `root` node at $t_0$, and proceeds by the model choosing APPLYRULE actions to generate the overall program structure from a closed set of grammar rules, then at leaves of the tree corresponding to variable terminals, the model switches to GENTOKEN actions to generate variables or constants from the open set. We describe this process in detail below.

### 3.1 APPLYRULE Actions

APPLYRULE actions generate program structure, expanding the current node (the *frontier node* at time step $t$: $n_{f_t}$) in a depth-first, left-to-right traversal of the tree. Given a fixed set of production rules, APPLYRULE chooses a rule $r$ from the subset that has a head matching the type of $n_{f_t}$, and uses $r$ to expand $n_{f_t}$ by appending all child nodes specified by the selected production. As an

example, in Fig. 1(b), the rule `Call` $\mapsto$ `expr`... expands the frontier node `Call` at time step $t_4$, and its three child nodes `expr`, `expr*` and `keyword*` are added to the derivation.

APPLYRULE actions grow the derivation AST by appending nodes. When a variable terminal node (e.g., `str`) is added to the derivation and becomes the frontier node, the grammar model then switches to GENTOKEN actions to populate the variable terminal with tokens.

**Unary Closure** Sometimes, generating an AST requires applying a chain of unary productions. For instance, it takes three time steps ($t_9 - t_{11}$) to generate the sub-structure `expr*` $\mapsto$ `expr` $\mapsto$ `Name` $\mapsto$ `str` in Fig. 1(a). This can be effectively reduced to one step of APPLYRULE action by taking the closure of the chain of unary productions and merging them into a single rule: `expr*` $\mapsto^*$ `str`. Unary closures reduce the number of actions needed, but would potentially increase the size of the grammar. In our experiments we tested our model both with and without unary closures (§ 5).

### 3.2 GENTOKEN Actions

Once we reach a frontier node $n_{f_t}$ that corresponds to a variable type (e.g., `str`), GENTOKEN actions are used to fill this node with values. For general-purpose PLs like Python, variables and constants have values with one or multiple tokens. For instance, a node that stores the name of a function (e.g., `sorted`) has a single token, while a node that denotes a string constant (e.g., `a='hello world'`) could have multiple tokens. Our model copes with both scenarios by firing GENTOKEN actions at one or more time steps. At each time

step, GENTOKEN appends one terminal token to the current frontier variable node. A special `</n>` token is used to "close" the node. The grammar model then proceeds to the new frontier node.

Terminal tokens can be generated from a pre-defined vocabulary, or be directly copied from the input NL. This is motivated by the observation that the input description often contains out-of-vocabulary (OOV) variable names or literal values that are directly used in the target code. For instance, in our running example the variable name `my_list` can be directly copied from the the input at $t_{12}$. We give implementation details in § 4.2.2.

## 4 Estimating Action Probabilities

We estimate action probabilities in Eq. (2) using attentional neural encoder-decoder models with an information flow structured by the syntax trees.

### 4.1 Encoder

For an NL description $x$ consisting of $n$ words $\{w_i\}_{i=1}^n$, the encoder computes a context sensitive embedding $\mathbf{h_i}$ for each $w_i$ using a bidirectional Long Short-Term Memory (LSTM) network (Hochreiter and Schmidhuber, 1997), similar to the setting in (Bahdanau et al., 2014). See supplementary materials for detailed equations.

### 4.2 Decoder

The decoder uses a RNN to model the sequential generation process of an AST defined as Eq. (2). Each action step in the grammar model naturally grounds to a time step in the decoder RNN. Therefore, the action sequence in Fig. 1(b) can be interpreted as unrolling RNN time steps, with solid arrows indicating RNN connections. The RNN maintains an internal state to track the generation process (§ 4.2.1), which will then be used to compute action probabilities $p(a_t|x, a_{<t})$ (§ 4.2.2).

#### 4.2.1 Tracking Generation States

Our implementation of the decoder resembles a vanilla LSTM, with additional neural connections (parent feeding, Fig. 1(b)) to reflect the topological structure of an AST. The decoder's internal hidden state at time step $t$, $\mathbf{s}_t$, is given by:

$$\mathbf{s}_t = f_{\mathrm{LSTM}}([\mathbf{a}_{t-1} : \mathbf{c}_t : \mathbf{p}_t : \mathbf{n}_{f_t}], \mathbf{s}_{t-1}), \quad (3)$$

where $f_{\mathrm{LSTM}}(\cdot)$ is the LSTM update function. [:] denotes vector concatenation. $\mathbf{s}_t$ will then be used to compute action probabilities $p(a_t|x, a_{<t})$ in Eq. (2). Here, $\mathbf{a}_{t-1}$ is the embedding of the previous action. $\mathbf{c}_t$ is a context vector retrieved from

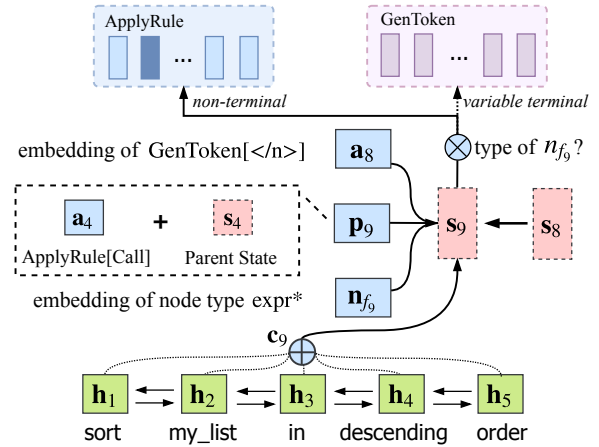

Figure 2: Illustration of a decoder time step ($t = 9$)

input encodings $\{\mathbf{h}_i\}$ via soft attention. $\mathbf{p}_t$ is a vector that encodes the information of the parent action. $\mathbf{n}_{f_t}$ denotes the node type embedding of the current frontier node $n_{f_t}$[3]. Intuitively, feeding the decoder the information of $n_{f_t}$ helps the model to keep track of the frontier node to expand.

**Action Embedding $\mathbf{a}_t$** We maintain two action embedding matrices, $\mathbf{W}_R$ and $\mathbf{W}_G$. Each row in $\mathbf{W}_R$ ($\mathbf{W}_G$) corresponds to an embedding vector for an action APPLYRULE[$r$] (GENTOKEN[$v$]).

**Context Vector $\mathbf{c}_t$** The decoder RNN uses soft attention to retrieve a context vector $\mathbf{c}_t$ from the input encodings $\{\mathbf{h}_i\}$ pertain to the prediction of the current action. We follow Bahdanau et al. (2014) and use a Deep Neural Network (DNN) with a single hidden layer to compute attention weights.

**Parent Feeding $\mathbf{p}_t$** Our decoder RNN uses additional neural connections to directly pass information from parent actions. For instance, when computing $\mathbf{s}_9$, the information from its parent action step $t_4$ will be used. Formally, we define the *parent action step* $p_t$ as the time step at which the frontier node $n_{f_t}$ is generated. As an example, for $t_9$, its parent action step $p_9$ is $t_4$, since $n_{f_9}$ is the node $\star$, which is generated at $t_4$ by the APPLYRULE[Call $\mapsto \dots$] action.

We model parent information $\mathbf{p}_t$ from two sources: (1) the hidden state of parent action $\mathbf{s}_{p_t}$, and (2) the embedding of parent action $\mathbf{a}_{p_t}$. $\mathbf{p}_t$ is the concatenation. The parent feeding schema enables the model to utilize the information of parent code segments to make more confident predictions. Similar approaches of injecting parent information were also explored in the SEQ2TREE model in Dong and Lapata (2016)[4].

---

[3] We maintain an embedding for each node type.

[4] SEQ2TREE generates tree-structured outputs by condi-

#### 4.2.2 Calculating Action Probabilities

In this section we explain how action probabilities $p(a_t|x, a_{<t})$ are computed based on $\mathbf{s}_t$.

**APPLYRULE** The probability of applying rule $r$ as the current action $a_t$ is given by a softmax[5]:

$$p(a_t = \text{APPLYRULE}[r]|x, a_{<t}) = \\ \text{softmax}(\mathbf{W}_R \cdot g(\mathbf{s}_t))^\intercal \cdot \mathbf{e}(r) \quad (4)$$

where $g(\cdot)$ is a non-linearity $\tanh(\mathbf{W} \cdot \mathbf{s}_t + \mathbf{b})$, and $\mathbf{e}(r)$ the one-hot vector for rule $r$.

**GENTOKEN** As in § 3.2, a token $v$ can be generated from a predefined vocabulary or copied from the input, defined as the marginal probability:

$$p(a_t = \text{GENTOKEN}[v]|x, a_{<t}) = \\ p(\text{gen}|x, a_{<t})p(v|\text{gen}, x, a_{<t}) \\ + p(\text{copy}|x, a_{<t})p(v|\text{copy}, x, a_{<t}).$$

The selection probabilities $p(\text{gen}|\cdot)$ and $p(\text{copy}|\cdot)$ are given by $\text{softmax}(\mathbf{W}_S \cdot \mathbf{s}_t)$. The probability of generating $v$ from the vocabulary, $p(v|\text{gen}, x, a_{<t})$, is defined similarly as Eq. (4), except that we use the embedding matrix of GENTOKEN, $\mathbf{W}_G$. To model the copy probability, we follow recent advances in modeling copying mechanism in neural networks (Gu et al., 2016; Jia and Liang, 2016; Ling et al., 2016), and use a pointer network (Vinyals et al., 2015) to compute the probability of copying the $i$-th word from the input by attending to input representations $\{\mathbf{h}_i\}$:

$$p(w_i|\text{copy}, x, a_{<t}) = \frac{\exp(\omega(\mathbf{h}_i, \mathbf{s}_t, \mathbf{c}_t))}{\sum_{i'=1}^n \exp(\omega(\mathbf{h}_{i'}, \mathbf{s}_t, \mathbf{c}_t))},$$

where $\omega(\cdot)$ is a DNN with a single hidden layer.

#### 4.3 Training and Inference

Given a dataset of pairs of NL descriptions $x_i$ and code snippets $c_i$, we parse $c_i$ into its AST $y_i$ and decompose $y_i$ into a sequence of oracle actions under the grammar model. The model is then optimized by maximizing the log-likelihood of the oracle action sequence. At inference time, we use beam search to approximate the best AST $\hat{y}$ in Eq. (1). See supplementary materials for the pseudo-code of the inference algorithm.

### 5 Experimental Evaluation

#### 5.1 Datasets and Metrics

**HEARTHSTONE** (HS) dataset (Ling et al., 2016) is a collection of Python classes that implement

---

| Dataset | HS | DJANGO | IFTTT |
|---|---|---|---|
| Train | 533 | 16,000 | 77,495 |
| Development | 66 | 1,000 | 5,171 |
| Test | 66 | 1,805 | 758 |
| Avg. tokens in description | 39.1 | 14.3 | 7.4 |
| Avg. characters in code | 360.3 | 41.1 | 62.2 |
| Avg. size of AST (# nodes) | 136.6 | 17.2 | 7.0 |
| *Statistics of Grammar* | | | |
| **w/o unary closure** | | | |
| # productions | 100 | 222 | 1009 |
| # node types | 61 | 96 | 828 |
| terminal vocabulary size | 1361 | 6733 | 0 |
| Avg. # actions per example | 173.4 | 20.3 | 5.0 |
| **w/ unary closure** | | | |
| # productions | 100 | 237 | – |
| # node types | 57 | 92 | – |
| Avg. # actions per example | 141.7 | 16.4 | – |

Table 2: Statistics of datasets and associated grammars

cards for the card game HearthStone. Each card comes with a set of fields (e.g., name, cost, and description), which we concatenate to create the input sequence. This dataset is relatively difficult: input descriptions are short, while the target code is in complex class structures, with each AST having 137 nodes on average.

**DJANGO** dataset (Oda et al., 2015) is a collection of lines of code from the Django web framework, each with a manually annotated NL description. Compared with the HS dataset where card implementations are somewhat homogenous, examples in DJANGO are more diverse, spanning a wide variety of real-world use cases like string manipulation, IO operations, and exception handling.

**IFTTT** dataset (Quirk et al., 2015) is a domain-specific benchmark that provides an interesting side comparison. Different from HS and DJANGO which are in a general-purpose PL, programs in IFTTT are written in a domain-specific language used by the IFTTT task automation App. Users of the App write simple instructions (e.g., `If Instagram.AnyNewPhotoByYou Then Dropbox.AddFileFromURL`) with NL descriptions (e.g., "*Autosave your Instagram photos to Dropbox*"). Each statement inside the `If` or `Then` clause consists of a channel (e.g., `Dropbox`) and a function (e.g., `AddFileFromURL`)[6]. This simple structure results in much more concise ASTs (7 nodes on average). Because all examples are created by ordinary Apps users, the dataset is highly noisy, with input NL very loosely con-

---

tioning on the hidden states of parent non-terminals, while our parent feeding uses the states of parent actions.

[5] We do not show bias terms for all softmax equations.

[6] Like Beltagy and Quirk (2016), we strip function parameters since they are mostly specific to users.

nected to target ASTs. The authors thus provide a high-quality filtered test set, where each example is verified by at least three annotators. We use this set for evaluation. Also note IFTTT's grammar has more productions (Tab. 2), but this does not imply that its grammar is more complex. This is because for HS and DJANGO terminal tokens are generated by GENTOKEN actions, but for IFTTT, all the code is generated directly by APPLYRULE actions.

**Metrics** As is standard in semantic parsing, we measure **accuracy**, the fraction of correctly generated examples. However, because generating an exact match for complex code structures is non-trivial, we follow Ling et al. (2016), and use token-level **BLEU-4** with as a secondary metric, defined as the averaged smoothed BLEU (Lin and Och, 2004) scores over all examples.

## 5.2 Setup

**Preprocessing** All input descriptions are tokenized using NLTK. We perform simple canonicalization for DJANGO, such as replacing quoted strings in the inputs with place holders. See supplementary materials for details. We extract unary closures whose frequency is larger than a threshold $k$ ($k = 30$ for HS and $50$ for DJANGO).

**Configuration** The size of all embeddings is 128, except for node type embeddings, which is 64. The dimensions of RNN states and hidden layers are 256 and 50, respectively. Since our datasets are relatively small for a data-hungry neural model, we impose strong regularization using recurrent dropouts (Gal and Ghahramani, 2016), together with standard dropout layers added to the inputs and outputs of the decoder RNN. We validate the dropout probability from $\{0, 0.2, 0.3, 0.4\}$. For decoding, we use a beam size of 15.

## 5.3 Results

Evaluation results for Python code generation tasks are listed in Tab. 3. Numbers for our systems are averaged over three runs. We compare primarily with two approaches: (1) Latent Predictor Network (LPN), a state-of-the-art sequence-to-sequence code generation model (Ling et al., 2016), and (2) SEQ2TREE, a neural semantic parsing model (Dong and Lapata, 2016). SEQ2TREE generates trees one node at a time, and the target grammar is not explicitly modeled a priori, but *implicitly* learned from data. We test both the original SEQ2TREE model released by the authors and our revised one (SEQ2TREE–UNK) that uses unknown word replacement to handle rare

|  | HS | | DJANGO | |
|---|---|---|---|---|
|  | ACC | BLEU | ACC | BLEU |
| Retrieval System[†] | 0.0 | 62.5 | 14.7 | 18.6 |
| Phrasal Statistical MT[†] | 0.0 | 34.1 | 31.5 | 47.6 |
| Hierarchical Statistical MT[†] | 0.0 | 43.2 | 9.5 | 35.9 |
| NMT | 1.5 | 60.4 | 45.1 | 63.4 |
| SEQ2TREE | 1.5 | 53.4 | 28.9 | 44.6 |
| SEQ2TREE–UNK | 13.6 | 62.8 | 39.4 | 58.2 |
| LPN[†] | 4.5 | 65.6 | 62.3 | 77.6 |
| Our system | 16.2 | **75.8** | 71.6 | **84.5** |
| Ablation Study |  |  |  |  |
| – frontier embed. | **16.7** | 75.8 | 70.7 | 83.8 |
| – parent feed. | 10.6 | 75.7 | 71.5 | 84.3 |
| – copy terminals | 3.0 | 65.7 | 32.3 | 61.7 |
| + unary closure | – | | 70.3 | 83.3 |
| – unary closure | 10.1 | 74.8 | – | |

Table 3: Results on two Python code generation tasks. [†]Results previously reported in Ling et al. (2016).

words (Luong et al., 2015). For completeness, we also compare with a strong neural machine translation (NMT) system (Neubig, 2015) using a standard encoder-decoder architecture with attention and unknown word replacement, and include numbers from other baselines used in Ling et al. (2016). On the HS dataset, which has relatively large ASTs, we use unary closure for our model and SEQ2TREE, and for DJANGO we do not.

**System Comparison** As in Tab. 3, our model registers 11.7% and 9.3% absolute improvements over LPN in accuracy on HS and DJANGO. This boost in performance strongly indicates the importance of modeling grammar in code generation. For the baselines, we find LPN outperforms others in most cases. We also note that SEQ2TREE achieves a decent accuracy of 13.6% on HS, which is due to the effect of unknown word replacement, since we only achieved 1.5% without it. A closer comparison with SEQ2TREE is insightful for understanding the advantage of our syntax-driven approach, since both SEQ2TREE and our system output ASTs: (1) SEQ2TREE predicts one node each time step, and requires additional "dummy" nodes to mark the boundary of a subtree. The sheer number of nodes in target ASTs makes the prediction process error-prone. In contrast, the APPLYRULE actions of our grammar model allows for generating multiple nodes at a single time step. Empirically, we found that in HS, SEQ2TREE takes more than 300 time steps on average to generate a target AST, while our model takes only 170 steps. (2) SEQ2TREE does not directly use productions in the grammar, which possibly leads to grammat-

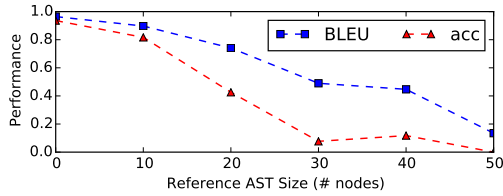

Figure 3: Performance w.r.t reference AST size on DJANGO

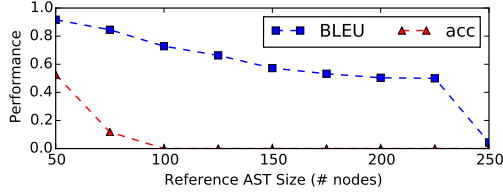

Figure 4: Performance w.r.t reference AST size on HS

ically incorrect ASTs and thus empty code outputs. We observe that the ratio of grammatically incorrect ASTs predicted by SEQ2TREE on HS and DJANGO are 21.2% and 10.9%, respectively, while our system guarantees grammaticality.

**Ablation Study** We also ablated our best-performing models to analyze the contribution of each component. "–frontier embed." removes the frontier node embedding $\mathbf{n}_{f_t}$ from the decoder RNN inputs (Eq. (3)). This yields worse results on DJANGO while gives slight improvements in accuracy on HS. This is probably because that the grammar of HS has fewer node types, and thus the RNN is able to keep track of $n_{f_t}$ without depending on its embedding. Next, "–parent feed." removes the parent feeding mechanism. The accuracy drops significantly on HS, with a marginal deterioration on DJANGO. This result is interesting because it suggests that parent feeding is more important when the ASTs are larger, which will be the case when handling more complicated code generation tasks like HS. Finally, removing the pointer network ("–copy terminals") in GENTO-KEN actions gives poor results, indicating that it is important to directly copy variable names and values from the input.

The results with and without unary closure demonstrate that, interestingly, it is effective on HS but not on DJANGO. We conjecture that this is because on HS it significantly reduces the number of actions from 173 to 142 (c.f., Tab. 2), with the number of productions in the grammar remaining unchanged. In contrast, DJANGO has a broader domain, and thus unary closure results in more productions in the grammar (237 for DJANGO vs. 100 for HS), increasing sparsity.

**Performance by the size of AST** We further investigate our model's performance w.r.t. the size

| | CHANNEL | FULL TREE |
|---|---|---|
| **Classical Methods** | | |
| posclass (Quirk et al., 2015) | 81.4 | 71.0 |
| LR (Beltagy and Quirk, 2016) | 88.8 | **82.5** |
| **Neural Network Methods** | | |
| NMT | 87.7 | 77.7 |
| NN (Beltagy and Quirk, 2016) | 88.0 | 74.3 |
| SEQ2TREE (Dong and Lapata, 2016) | 89.7 | 78.4 |
| Doubly-Recurrent NN (Alvarez-Melis and Jaakkola, 2017) | **90.1** | 78.2 |
| Our system | 90.0 | 82.0 |
| – parent feed. | 89.9 | 81.1 |
| – frontier embed. | **90.1** | 78.7 |

Table 4: Results on the noise-filtered IFTTT test set of ">3 agree with gold annotations" (averaged over three runs), our model performs competitively among neural models.

of the gold-standard ASTs in Figs. 3 and 4. Not surprisingly, the performance drops when the size of the reference ASTs increases. Additionally, on the HS dataset, the BLEU score still remains at around 50 even when the size of ASTs grows to 200, indicating that our proposed syntax-driven approach is robust for long code segments.

**Domain Specific Code Generation** Although this is not the focus of our work, evaluation on IFTTT brings us closer to a standard semantic parsing setting, which helps to investigate similarities and differences between generation of more complicated general-purpose code and and more limited-domain simpler code. Tab. 4 shows the results, following the evaluation protocol in (Beltagy and Quirk, 2016) for accuracies at both channel and full parse tree (channel + function) levels. Our full model performs on par with existing neural network-based methods, while outperforming other neural models in full tree accuracy (82.0%). This score is close to the best classical method (LR), which is based on a logistic regression model with rich hand-engineered features (e.g., brown clusters and paraphrase). Also note that the performance between NMT and other neural models is much closer compared with the results in Tab. 3. This suggests that general-purpose code generation is more challenging than the simpler IFTTT setting, and therefore modeling structural information is more helpful.

**Case Studies** Finally, we present output examples in Tab. 5. On HS, we observe that most of the time our model gives correct predictions by filling learned code templates from training data with arguments (e.g., cost) copied from input. However, we do find interesting examples indicating that the model learns to generalize beyond trivial copy-

**input** *<name> Brawl </name> <cost> 5 </cost> <desc> Destroy all minions except one (chosen randomly) </desc> <rarity> Epic </rarity> ...*

**pred.**
```python
class Brawl(SpellCard):
    def __init__(self):
        super().__init__('Brawl', 5, CHARACTER_CLASS.
            WARRIOR, CARD_RARITY.EPIC)
    def use(self, player, game):
        super().use(player, game)
        targets = copy.copy(game.other_player.minions)
        targets.extend(player.minions)
        for minion in targets:
            minion.die(self)
```
A

**ref.**
```python
minions = copy.copy(player.minions)
minions.extend(game.other_player.minions)
if len(minions) > 1:
    survivor = game.random_choice(minions)
    for minion in minions:
        if minion is not survivor: minion.die(self)
```
B

**input** *join app_config.path and string 'locale' into a file path, substitute it for localedir.*

**pred.**
```python
localedir = os.path.join(
    app_config.path, 'locale') ✓
```

**input** *self.plural is an lambda function with an argument n, which returns result of boolean expression n not equal to integer 1*

**pred.**
```python
self.plural = lambda n: len(n) ✗
```
**ref.**
```python
self.plural = lambda n: int(n!=1)
```

Table 5: Predicted examples from HS (1st) and DJANGO. Copied contents (copy probability > 0.9) are highlighted.

ing. For instance, the first example is one that our model predicted wrong — it generated code block A instead of the gold B (it also missed a function definition not shown here). However, we find that the block A actually conveys part of the input intent by destroying all, not some, of the minions. Since we are unable to find code block A in the training data, it is clear that the model has learned to generalize to some extent from multiple training card examples with similar semantics or structure.

The next two examples are from DJANGO. The first one shows that the model learns the usage of common API calls (e.g., os.path.join), and how to populate the arguments by copying from inputs. The second example illustrates the difficulty of generating code with complex nested structures like lambda functions, a scenario worth further investigation in future studies. More examples are attached in supplementary materials.

## 6 Related Work

**Code Generation and Analysis** Most existing works on code generation focus on generating code for domain specific languages (DSLs) (Kushman and Barzilay, 2013; Raza et al., 2015; Manshadi et al., 2013), with neural network-based approaches recently explored (Parisotto et al., 2016; Balog et al., 2016). For general-purpose code gen-

eration, besides the general framework of Ling et al. (2016), existing methods often use language and task-specific rules and strategies (Lei et al., 2013; Raghothaman et al., 2016). A similar line is to use NL queries for code retrieval (Wei et al., 2015; Allamanis et al., 2015). The reverse task of generating NL summaries from source code has also been explored (Oda et al., 2015; Iyer et al., 2016). Finally, there are probabilistic models of source code (Maddison and Tarlow, 2014; Nguyen et al., 2013). The most relevant work is Allamanis et al. (2015), which uses a factorized model to measure semantic relatedness between NL and ASTs for code retrieval, while our model tackles the more challenging generation task.

**Semantic Parsing** Our work is related to the general topic of semantic parsing, where the target logical forms can be viewed as DSLs. The parsing process is often guided by grammatical formalisms like combinatory categorical grammars (Kwiatkowski et al., 2013; Artzi et al., 2015), dependency-based syntax (Liang et al., 2011; Pasupat and Liang, 2015) or task-specific formalisms (Clarke et al., 2010; Yih et al., 2015; Krishnamurthy et al., 2016; Misra et al., 2015; Mei et al., 2016). Recently, there are efforts in designing neural network-based semantic parsers (Misra and Artzi, 2016; Dong and Lapata, 2016; Neelakantan et al., 2016). Several approaches have be proposed to utilize grammar knowledge in a neural parser, such as augmenting the training data by generating examples guided by the grammar (Kociský et al., 2016; Jia and Liang, 2016). Liang et al. (2016) used a neural decoder which constrains the space of next valid tokens in the query language for question answering. Finally, the structured prediction approach proposed by Xiao et al. (2016) is closely related to our model in using the underlying grammar as prior knowledge to constrain the generation process of derivation trees, while our method is based on a unified grammar model which jointly captures production rule application and terminal symbol generation, and scales to general purpose code generation tasks.

## 7 Conclusion

This paper proposes a syntax-driven neural code generation approach that generates an abstract syntax tree by sequentially applying actions from a grammar model. Experiments on both code generation and semantic parsing tasks demonstrate the effectiveness of our proposed approach.

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
