# Peer review of "A Syntactic Neural Model for General-Purpose Code Generation"

_ACL 2017 — decision unknown_

[Official Review · Reviewer 1 · rating 4 · confidence 4]
soundness 5 · originality 3 · clarity 5 · impact 3 · substance 5 · appropriateness 5 · meaningful comparison 4 · presentation format Oral Presentation

Summary: The paper proposes a neural model for predicting Python syntax trees
from text descriptions. Guided by the actual Python grammar, the model
generates tree nodes sequentially in a depth-first fashion. Key ideas include
injecting the information from the parent node as part of the LSTM input, a
pointer network for copying the terminals, and unary closure which collapses
chains of unary productions to reduce the tree size. The model is evaluated on
three datasets from different domains and outperforms almost all previous work.

Strengths:

The paper is overall very well-written. The explanation of system is clear, and
the analysis is thorough.

The system itself is a natural extension of various ideas. The most similar
work include tree-based generation with parent feeding (Dong and Lapata, 2016)
and various RNN-based semantic parsing with copy mechanism (Jia and
Liang, 2016; Ling et al., 2016). [The guidance of parsing based on grammar is
also explored in Chen Liang et al., 2016 (https://arxiv.org/abs/1611.00020)
where a code-assist system is used to ensure that the code
is valid.] Nevertheless, the model is this paper stands out as it is able to
generate much longer and more complex programs than most previous work
mentioned. 

Weaknesses:

The evaluation is done on code accuracy (exact match) and BLEU score. These
metrics (especially BLEU) might not be the best metrics for evaluating the
correctness of programs. For instance, the first example in Table 5 shows that
while the first two lines in boxes A and B are different, they have the same
semantics. Another example is that variable names can be different. Evaluation
based on what the code does (e.g., using test cases or static code analysis)
would be more convincing.

Another point about evaluation: other systems (e.g., NMT baseline) may generate
code with syntactic error. Would it be possible to include the result on the
highest-scoring well-formed code (e.g., using beam search) that these baseline
systems generate? This would give a fairer comparison since these system can
choose to prune malformed code.

General Discussion:

* Lines 120-121: some approaches that use domain-specific languages were also
guided by a grammar. One example is Berant and Liang, 2014, which uses a pretty
limited grammar for logical forms (Table 1). In addition to comparing to that
line of work, emphasizing that the grammar in this paper is much larger than
most previous work would make this work stronger.

* Lines 389-397: For the parent feeding mechanism, is the child index being
used? In other words, is p_t different when generating a first child versus a
second child? In Seq2Tree (Dong and Lapata, 2016) the two non-terminals would
have different hidden states.

* Line 373: Are the possible tokens embedded? Is it assumed that the set of
possible tokens is known beforehand?

* The examples in the appendix are nice.

---

I have read the author response.

[Official Review · Reviewer 2 · rating 4 · confidence 4]
soundness 5 · originality 3 · clarity 4 · impact 3 · substance 4 · appropriateness 5 · meaningful comparison 4 · presentation format Oral Presentation

This paper presents a method for translating natural language descriptions into
source code via a model constrained by the grammar of the programming language
of the source code.  I liked this paper - it's well written, addresses a hard
and interesting problem by taking advantage of inherent constraints, and shows
significant performance improvements. 

Strengths:
- Addresses an interesting and important problem space. 
- Constraints inherent to the output space are incorporated well into the
model. 
- Good evaluation and comparisons; also showing how the different aspects of
the model impact performance.
- Clearly written paper.

Weaknesses:
- My primary and only major issue with the paper is the evaluation metrics.
While accuracy and BLEU4 are easy to compute, I don't think they give a
sufficiently complete picture.                          Accuracy can easily miss
correctly
generated
code because of trivial (and for program functionality, inconsequential)
changes.  You could get 0% accuracy with 100% functional correctness.  As for
BLEU, I'm not sure how well it evaluates code where you can perform significant
changes (e.g., tree transformations of the AST) without changing functionality.
 I understand why BLEU is being used, but it seems to me to be particularly
problematic given its token level n-gram evaluation.  Perhaps BLEU can be
applied to the ASTs of both reference code and generated code after some level
of normalization of the ASTs?  What I would really like to see is an evaluation
testing for functional equivalence of reference and generated code. 
Understandably this is difficult since test code will have to be written for
each reference.  However, even if this is done for a random (reasonably small)
subsample of the datasets, I think it would give a much more meaningful
picture. 

Minor issues:
- Page 2, para 2: "structural information helps to model information flow
within the network": by "network", do you mean the AST?

- Section 4.2.1, Action Embedding: Are the action embedding vectors in W_R and
W_G simply one-hot vectors or do you actually have a non-trivial embedding for
the actions?  If so, how is it computed?  If not, what is the difference
between the vectors of W_R and e(r) in equation 4?

- Section 5.2, Preprocessing:  If you replace quoted strings in the
descriptions for the DJANGO dataset, how are cases where those strings need to
be copied into the generated code handled?  It is also mentioned (in the
supplementary material) that infrequent words are filtered out.  If so, how do
you handles cases where those words describe the variable name or a literal
that needs to be in the code?

I have read the author response.

[Official Review · Reviewer 3 · rating 4 · confidence 4]
soundness 5 · originality 3 · clarity 5 · impact 3 · substance 5 · appropriateness 5 · meaningful comparison 4 · presentation format Oral Presentation

- Strengths:

The approach proposed in the paper seems reasonable, and the experimental
results make the approach seem promising. There are two features of the 
approach. One feature is that the approach is for general-purpose programming
languages. It might be applicable to Java, C++, etc. However, proof 
is still needed. Another feature is its data-driven syntactic neural model,
which is described in Section 3 (together with Section 4 I think). 
By the neural model, it brings around 10% improvement over another same-purpose
approach LPN in accuracy (according to the experimental data). 
Overall, this is nice work with clear motivation, methodology, data analysis,
and well-organized presentation.

- Weaknesses:

1. At Line 110, the authors mentioned hypothesis space. I did not know what it
means until I read the supplementary materials. Because such materials 
will not be included in the full paper, in my opinion it is better to give some
explanation on hypothesis space. 

2. Section 3 introduces the grammar model and Section 4 describes Action
probability estimation. My understanding is that the latter is a part of the
former. The two section titles do not reflect this relation. At least Section 3
does not explain all about the grammar model. 

3. About the experimental data, I'm wondering how the authors train their model
before doing the experiments. How many datasets are used. Is it true that 
more the model get trained, more accuracy can be obtained?  How about the
efficiency of the two approaches, the one in the paper and LPN?   

4. Are there differences between the neural network-based approaches that are
used for code generation of general-purpose language and those of domain
specific ones? 

5. The authors claim that their approach scale up to generation of complex
programs. I did not find any argument in the paper to backup this conclusion. 

Minor comments:

Line 117: The underlying syntax => the syntax of which language? (NL or PL?)
Line 148: Are there any constraints on x? 
Line 327: The decoder uses a RNN => The decoder uses an RNN?
Reference:  format is inconsistent

- General Discussion:

This paper proposes a data-driven syntax-based neural network model for code
generation in general-purpose programming langauge, i.e., Python. 
The main idea of the approach is first to generate a best-possible AST using a
probabilistic grammar model for a given statement in natural language, and
then ecode AST into surce code using deterministic generation tools. Generating
code from an AST is relatively easy. The key point is the first step. 
Experimental results provided in the paper show the proposed approach
outperform some other state-of-the-art approaches.